
# Using Formvar to Capture Ice Crystals and Retrieve Roughness Parameters

Omer Celebi[1], Andrew R.D. Smedley[1], Paul Connolly[1] Ann R. Webb[1]

[1]Department of Earth and Environmental Sciences, University of Manchester, Manchester, M13 9PL, UK

*Correspondence to*: Omer Celebi (omer.celebi@postgrad.manchester.ac.uk)

**Abstract.** Ice crystal roughness is a poorly observed and understood parameter, yet it significantly influences crystal's scattering properties and consequently impacts radiative transfer in the atmosphere, contributing to uncertainties in weather and climate forecasting. In this study, we introduce a novel approach to obtain high-resolution roughness measurements, building on the traditional formvar method for capturing ice crystals. Ice crystals were grown in the Manchester Ice Cloud
Chamber, collected, and subsequently imaged using various techniques, including a scanning optical profilometer, which enabled the identification of roughness features as small as 0.8 µm. This approach provides critical insights into roughness characteristics that are significant for improving radiative transfer models and forecasts.

## 1 Introduction

The mean equilibrium surface temperature of Earth is significantly affected by clouds that are responsible for both trapping
terrestrial radiation as well as reflecting solar radiation back to space. The balance between these processes determines the change in surface temperature due to changes in the cloud/ presence of the cloud (Hong Y, 2015). As indicated in the IPCC report (2013), the sensitivity in the prediction models for climate changes can reach up to $1.5°C–2.0°C$. The reason for this range of estimates in the prediction of surface temperature was noted in the report as primarily being due to our limited knowledge of cloud feedback ((IPCC), 2013). This lack of understanding of the physical properties of clouds is particularly
acute for ice clouds since they are complex over a range of spatial scales and cover large portions of the globe, thus causing significant errors in predictions of climate change. These uncertainties have been substantially mitigated through rigorous, collaborative efforts by the research community, dedicated to advancing the understanding and precise quantification of cloud feedback mechanisms across a range of cloud regimes ((IPCC), 2023). As the scientific community has worked to address these uncertainties, our focus here is on issues that can be further mitigated through a deeper understanding of the
microscopic properties of ice crystals. This comprises backscattering properties of ice crystals based on their microphysical properties, including shape, orientation, complexity, and roughness.

Cloud models can give accurate results if the inputs represent the actual conditions. Therefore, there have been many efforts over the last 80 years (Bailey, et al., 2009) to understand ice crystal habits formation as a function of environmental



conditions, enabling the input to cloud models to be as accurate as possible. In one of the earliest systematic investigations, Magono and Chung (1966) developed a comprehensive classification of natural snow crystal types, creating a foundational framework that has since guided research in snowflake morphology. More recently, in 2005, an experiment was conducted by Libbrecht to explain the mechanism that creates a general hexagonal prism shape for ice crystal growth when their sizes are between 10-100 µm (Libbrecht, 2005). Also, in the same year, Connolly et al. (2005) studied the correlation between the

electric field and aggregation of ice crystals and also made observations of chains of ice crystals during fieldwork into deep convective outflows. The results of the study also compared in many aspects with the work done by Saunders and Wahab (1975) since their field data were similar.

As the habit of the ice crystal varies under different conditions, its contribution to scattering behaviour and overall cloud

reflection capacity varies. For example, in 2015, Smith et al. investigated scattering by hollow columnar crystals. These types of column crystals have hollows in their end faces that alter the scattering behaviour. It was found that the phase functions are not affected too much, but there is a difference in asymmetry parameters as a result of different reflections occurring around the hollows (Smith, et al., 2015).

Throughout these investigations, the methods used to analyse and observe the ice crystals also developed over time as technology developed. Initially, ice crystals were able to be captured and analysed with a solution often referred to as 'formvar'. Formvar has been widely utilized across numerous fields for many years, showcasing its adaptability and broad range of applications, including providing sufficient detail to capture surface roughness accurately. In general, it has been coupled with electron scanning microscopy. Kahler et al. (1951)used formvar to coat the copper screens for visualization of

copper-cystine fibres under electron microscopy. Also, it has been applied in the biological sciences.  For example, more recently, in the treatment of human tumours, Corona-Ortega et al. (2018) used a formvar solution to analyse human tissue under transmission electron microscopy (TEM), whilst Wang et al. (2020) used a formvar-like epoxy resin solution to encapsulate steel pipe structures to study their damping properties. Formvar solutions have also found a place in the field of nanotechnology. Auchter et al. (2018) applied them in the production process of membranes to add strength whilst thinning

the material created. These membranes are used at micro- and nanoscales to create filtration systems, pressure sensors, and the like (Auchter, et al., 2018). Another study conducted was by Hamacher-Barth et al (2013) who used formvar films to size atmospheric particles by collecting and then scanning them under an electron microscope, noting it was useful for those particles with size under a micrometre.

Building on its broad applicability in various fields, the general usage of formvar in the field of atmospheric science was first described in 1941 by Schaefer as the solution of polyvinyl formal resin (formvar) dissolved in ethylene dichloride (Schaefer, 1941). Briefly, this solution is used to capture ice crystals and create its formvar replica by allowing the solution to flow around the crystal, and as the solvent evaporates, it leaves behind a plastic replica formed of the solid formvar solute.





Rucklidge (1965) conducted an experiment that involved the usage of the formvar solution to create replicas and examine them under an electron microscope. The samples were taken from cloud chambers. Size and shape properties were also studied in this work. The work of Griggs and Jayaratne (1986) led to improvements in the capture quality, including precautions during the application of the procedure. Since then, the method has frequently been used to create and then image replicas, sometimes using high-resolution optical microscopy, as in Smith et al (2015) who used this technique classify and explain the habit of the crystals grown in different conditions. Finnegan and Pitter (1988) investigated the replica process to explain aggregation and secondary ice formation. In the study of Miloshevich et al. (1996), formvar was used to collect cloud particles as a part of the process for a balloon-borne cloud particle replicator to measure the vertical profile of particles. Another replicator was designed and used by Warburton et al. (1983) for the study of transmission of snowfall. This replicator also included a formvar to capture crystals by creating a thin film and allowing crystals to impact it to produce detailed imprints on the film.

In the field of atmospheric science there is also a history of using of formvar. As these studies in atmospheric science demonstrate, whilst the use of formvar was fairly widespread towards the end of the 20th century, the replicas were only used for imaging of the crystals and habit classification but not to retrieve any finer details like roughness. In part, this may be attributed to a move towards optical imaging probes that offer automatic sizing and shape determination but at a lower resolution than that available from standard optical imaging of formvar replicas. These probes provide fast and in-situ imaging, but the resolution of the images is not adequate to analyse the surface details or roughness.

However, work in other fields suggests it should be possible to capture fine details of ice crystals to retrieve roughness values. There is a pioneer study that investigates the formvar replica process and its ability to capture roughness details (Agar, et al., 1956). This study used formvar solution to coat metal alloys, and an interferometer microscope was used to conduct further measurements. Ideally, a uniform thickness of formvar should be present across the substrate. In order to provide this, a method similar to the one used by Revell et al. (1955) was applied, whereby the sample was submerged into a formvar solution before being drained. On average, the thickness of the formvar solution achieved was around 700 Å. After the coating process, scans of the reverse of the replica and the original material were compared, and the difference between them was found to be negligible.

Up to this point, it has not been possible to get a direct measurement of the roughness of ice crystals in order to implement it into scattering models, though indirect means have been used to provide roughness parameters. For example, in Collier et al., a sand particle was scanned with an Atomic Force Microscope (AFM) and its roughness parameters were used in models as a proxy (Collier, et al., 2016). Also, Riskila et al. (2021) investigated the scattering of ice crystals by placing a finite, thin, rough element on an infinitely large vacuum boundary. All these works stated the significant effect of roughness on scattering behaviour of ice crystals, and in turn the overall contribution of clouds to the climate.



To address these issues and move beyond having to use proxies of roughness, in this study, we demonstrate a technique to obtain a direct measurement of the roughness of ice crystals by coupling the relatively old method of capturing ice crystals in formvar with newer instrumentation in the form of an optical profilometer. Nevertheless, using a such an old technique requires some modifications to improve the quality of capture and allow for detailed imaging that then enables a direct measurement of the roughness and provides a useful complementary technique to the current suite of cloud imaging probes. Roughness values smaller than 100 nm will not be considered, as they are unlikely to influence scattering for wavelengths less than 1000 nm.





## 2 Methodology

The formvar replica technique has traditionally been an efficient way of capturing ice crystals and preserving their surface structure, which is particularly important for ice crystals which would otherwise melt once collected and brought to laboratory temperature. While it has successfully been used for determining crystal habits and features such as hollowness,

identifying roughness of a crystal requires evidence that the formvar can accurately reproduce the necessary high-resolution detail. We present the final methodology developed to determine the scale of roughness of ice crystals. The results section will also illustrate some of the intervening stages in developing the technique, for example the imaging of salt crystals as proof of concept that the formvar replicas could maintain structure of the magnitude expected of ice crystal roughness.

The initial step of the traditional method is the placement of a solution of 0.6wt% polyvinyl formal resin in ethylene dichloride by using a brush onto a glass slide that is then used to capture the ice crystals. Once the formvar solution has subsequently dried out, a replica of the ice crystal is left in the solid formvar which can be investigated under microscopy to retrieve features of the crystal.

The brush leaves a thin layer of solution after it is applied to the slide. This has the advantage of drying rapidly but the crystals cannot be covered fully. This results in imprints of crystals which are suitable for habit identification but not ideal for roughness detection. A more effective approach involves capturing the entire crystal by suspending and encasing it in formvar, thereby ensuring that all surface cavities remain filled during the solidification of the solution and the sublimation or ablation of the crystal. As most of the crystals generated in the Manchester Ice Cloud Chamber (MICC) fall under 1 mm

in size, that depth of solution was targeted for the experiment, but brings its own challenges.

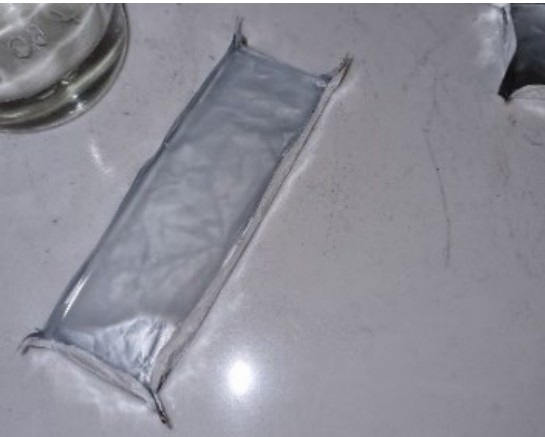

**Figure 1.** Aluminium foil container with formvar solution pool.





As shown in Figure 1, to achieve the increased thickness of formvar solution required, aluminium foil was used to create a wall around the glass slide, creating a pool of the solution to a desired level of thickness. It is critical to have enough depth for ice crystals to submerge fully but the evaporation of the chloroform solvent from a thicker layer of formvar, at the low temperatures required for crystal formation, proved too slow at standard atmospheric pressure. Early trials found that the chloroform could not completely evaporate before the crystals deformed or ablated.


The study of Kawamura et al (1987). shows, the evaporation rate per unit area is calculated following Eq. (1):

$$E = k * M * P(T_s)/RT \tag{1}$$

Where $k$ is the mass transfer coefficient, $M$ is the molecular weight, $P(T_s)$ is the vapour pressure at $T_s$ which is the temperature of solution surface, while $R$ and $T$ are the gas constant and absolute temperature of air respectively. The

evaporation rate is directly proportional to vapour pressure, thus at a given temperature increasing the vapour pressure increases the evaporation rate. As seen in Figure 2 at a temperature of –20°C, the vapour pressure of chloroform is decreased by a factor of at least 5.5 when compared to room temperature.

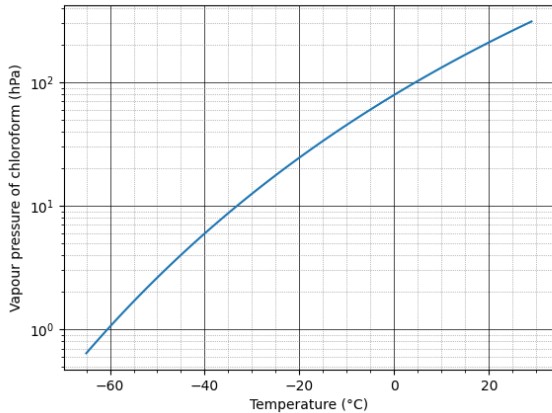

**Figure 2.** Vapour pressure of chloroform (hPa) vs temperature (°C) adapted from Cheric (1995).

This suggests a solution to the problem caused by slow evaporation of chloroform: that by reducing the external pressure on the slide the evaporation rate would be increased. Accordingly, the formvar slide, with its captured ice crystals was sealed inside a bell jar and air slowly pumped from within. The target pressure was 140–200 hPa achieved at a flowrate (set to 10

$m^3min^{-1}$) that would not disturb the sample contained within the bell jar.





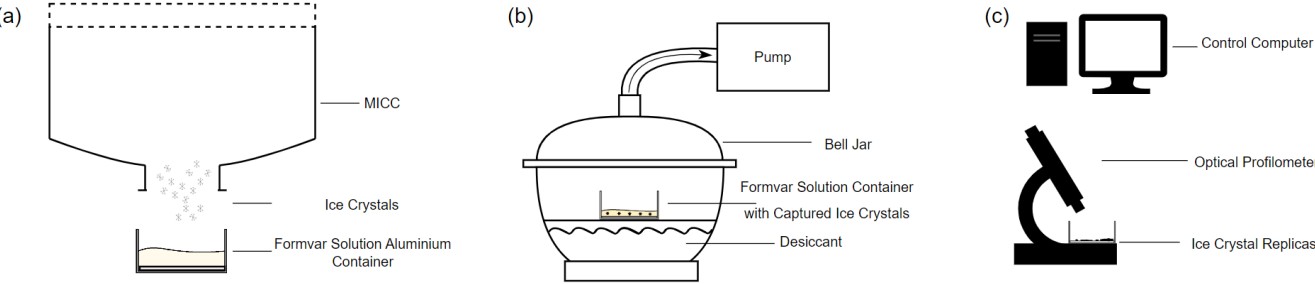

**Figure 3.** Schematic (not to scale) showing steps described in methodology: **(a)** sample collection, **(b)** evaporation process, and **(c)** optical profilometer scanning.

At this flowrate, it took a few minutes to reach the desired pressure level, and once it was reached, the evaporation visually inspected: as the evaporation occurs, the colour of the solution goes from transparent to silky white. The sample was left to evaporate for 10 minutes to ensure the process was complete.

Samples were initially examined using an optical profilometer, specifically the Keyence X200K 3D Laser Microscope, to ensure that the crystal replicas remained intact without undergoing any additional deformations. This preliminary inspection enabled the identification of areas of interest for more detailed analysis. Once selected, these areas were thoroughly surface-scanned using the same instrument, and the resulting scan data was stored for further evaluation. The primary advantage of utilizing the optical profilometer over earlier imaging techniques lies in its dual functionality. It offers a rapid initial optical assessment, followed by a highly detailed 3D surface scan that captures intricate topographical features with precision. The optical profilometer, such as a laser scanning confocal microscope, utilizes non-contact techniques to measure surface profiles, roughness, and thickness across different materials. It employs both laser and white light sources to simultaneously capture laser intensity, colour, and height data, allowing for the generation of fully focused colour images and detailed 3D height profiles. In contrast to traditional methods like scanning electron microscopy, which may distort samples due to electron beam interaction, the profilometer maintains sample integrity while delivering high-resolution data. This capability makes it well-suited for both quick assessments and comprehensive surface analysis, providing excellent versatility and accuracy, compared to other methods used in the early stages of technique development. After the scanning data was retrieved, it was examined and analysed using of software package called Gwyddion. This program allows users to import surface scanning data and to focus on extracting detailed data from a specified area of the scanned surface, in this case the crystal replicas and not the surrounding formvar, though background areas of formvar were scanned to provide a baseline roughness value. It enables parameters like mean roughness to be calculated following Eq. (2):

$$R_a = \frac{1}{L} \int_0^L |Z(x)| dx \tag{2}$$





Where L is evaluation length and Z(x) is height profile (Gadelmawla, et al., 2002) and root mean square (RMS) roughness
calculated following Eq. (3):

$$R_q = \sqrt{\frac{1}{L}\int_0^L Z(x)^2 dx} \tag{3}$$

## 3. Results and Discussion

The methodology outlined in the previous section was developed to demonstrate that it is possible to extract roughness information directly from ice crystals formed within clouds. It was inspired by the observation that traditional optical
microscope images are not sufficiently detailed to provide information on roughness as the required high image resolution cannot be achieved.

During the initial examination of the optical microscope sample images retrieved with using the brush technique, there were features observed that could be artefacts of the formvar process itself, rather than features of the original crystals.  Clarifying
whether these are artefacts or belong to the crystals themselves, however, requires high-resolution images. Initially, scanning electron microscopy (SEM) was employed to capture the high-resolution images necessary for the analysis by scanning the surface of the material with a focused beam of electrons. SEM was chosen for its ability to achieve resolutions of up to approximately 100 nm, which theoretically provides sufficient detail to meet the 100nm roughness measurement goal. This high level of resolution made SEM an ideal tool for detailed surface characterization in the early stages of the study,
allowing for precise visualization of microstructural features.

In order to explore this further, small areas of interest were chosen from the optical microscope examination and prepared for SEM imaging. Once each area is chosen, the whole slide was covered with foil, leaving a small hole approximately ~1mm$^2$ in size in the foil which defines the area of interest (Figure 4). Since scanning might need to be repeated several times to get
images with different resolutions and to prevent causing damage to the whole slide, only a small portion of a slide was prepared for imaging by coating with gold, and the scanning then carried out to image the selected feature.





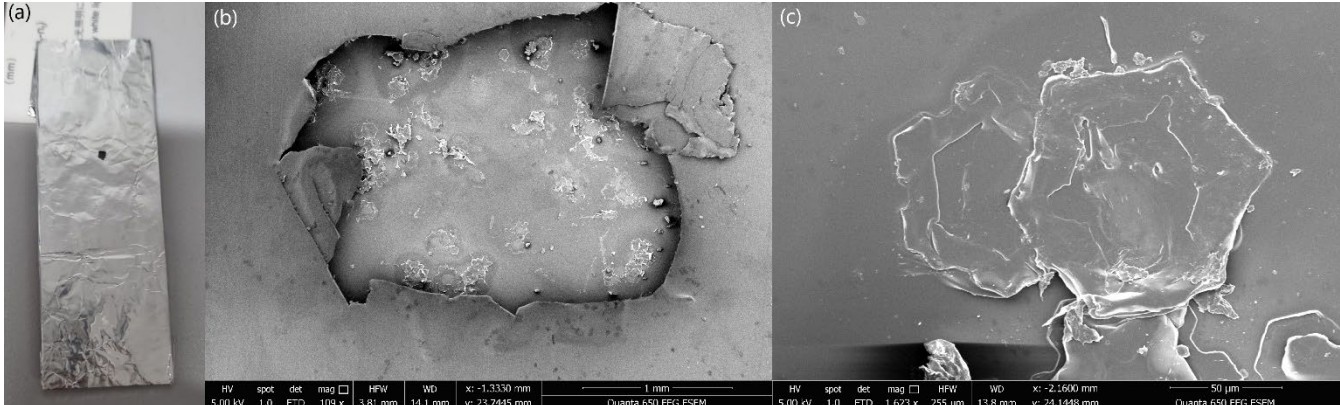

**Figure 4.** SEM imaging steps described: **(a)** sample 73 covered with foil with an opening on it, **(b)** the selected area's SEM image and **(c)** close-up image of 2 hexagonal plates that sit on each other.

In total, eight samples were prepared for this preliminary investigation, and were imaged with varying zoom scales during three different SEM sessions. When those images were examined thoroughly, it can be seen that a significant portion of the features are artefacts of the formvar process. It was thought possible that the formvar material might start to deform after or whilst the ice crystals evaporate. Another critical observation from these initial SEM images is that the formvar may not be covering the crystal completely due to its density and crystal size. It was often observed that crystals did not sink completely into the formvar under some conditions (Figure 5) leaving an opening in the upper surface of the replica. This is mainly attributed to insufficient thickness of the formvar layer, though the density and viscosity may have been contributing factors preventing the crystal from sinking. A relatively simple workaround for this problem is to increase the depth of the formvar solution layer by constructing a container, though it has consequences in terms of the evaporation rate as discussed previously.

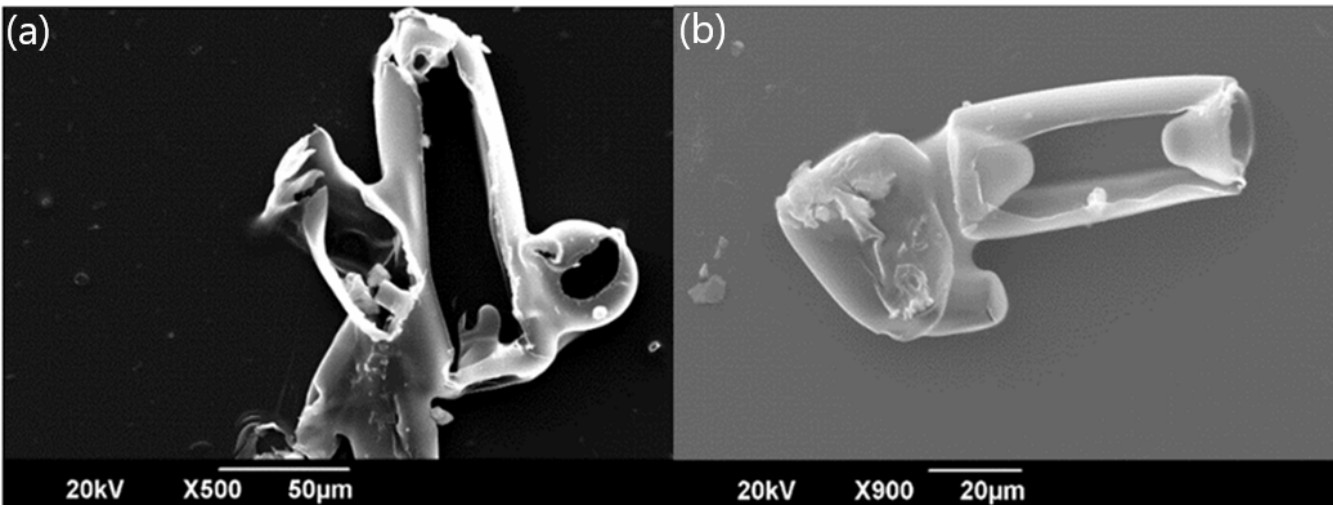





**Figure 5.** SEM images of the ice crystals which did not sink fully into the formvar solution and consequently each replicas exhibits a hole in its upper surface.

### 3.1 Validation using salt crystals

Though the procedure described in Section 2 allowed us to measure the roughness of atmospheric ice crystals, an essential step is to ensure that the roughness values obtained from a replica are the same as would be measured from the crystal itself. To this end, the process was repeated with table salt (sodium chloride) crystals as a validation source, though with slight adaptations.

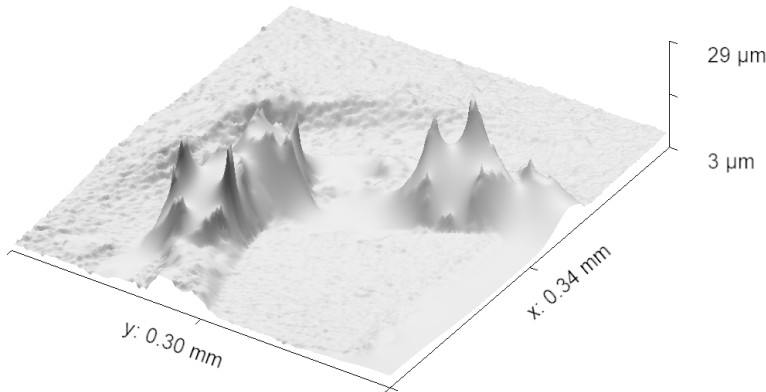

**Figure 6.** Surface scanning of replica of salt crystal created by using brush technique.

As with ice crystals, once the chloroform had evaporated from the formvar solution, the formvar leaves behind a replica of the salt crystals. At this stage though, the salt crystals remain, and so must be removed by being dissolved in water. For this the sample slide was gently placed in a water bath and left for 15 minutes until the salt dissolved and could be washed away. With the salt removed from the sample, only the replica shell remained, and it was scanned with the SEM microscope. In addition, salt crystals from the same batch were placed directly on a microscope slide and imaged and analysed in the same manner.



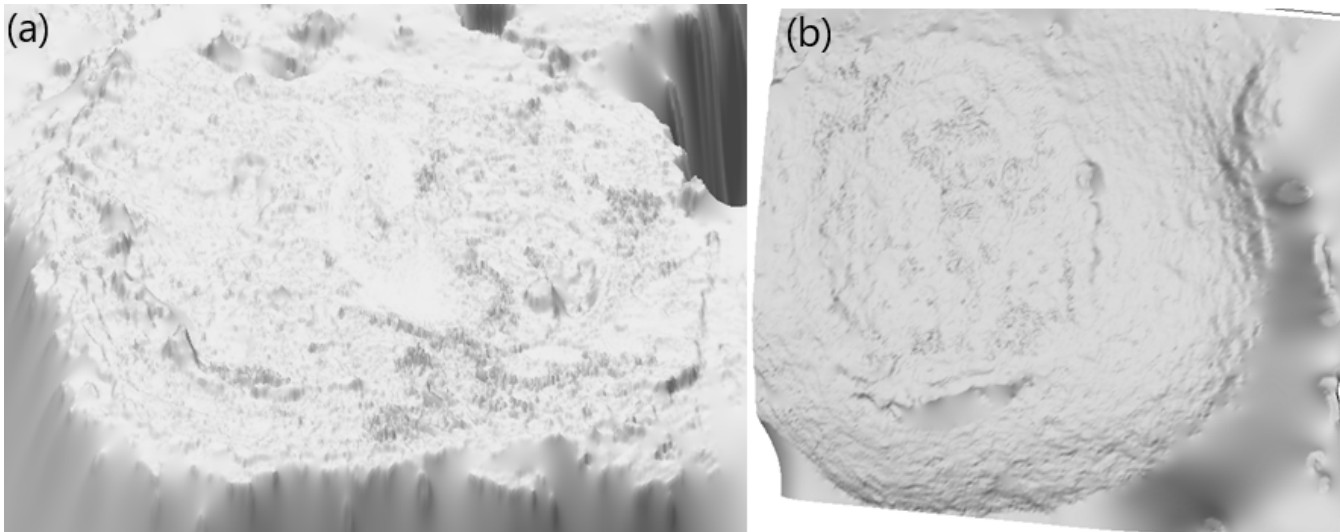


**Figure 7. (a)** Scan of randomly chosen salt crystal without formvar solution application. **(b)** Scan of randomly chosen formvar shell after salt crystal removal.

The image in Figure 7 (left), shows the result scanning of a sample salt crystal without the application of formvar whereas

the righthand side of the figure shows the result of the replica formvar process. In a comparison of both images, it can be seen that a sufficient depth of formvar allows the full surface of the crystal to be imaged and complete scan of the surface was retrieved. The results of roughness analysis are shown in Table 1.

From these, we can see that the formvar replica process retains the roughness parameter measurement of the original crystal

with a good level of accuracy. The deviation between the two samples is less than 5% which gives confidence that the measurement technique can represent true roughness parameters when it is applied to ice crystals. It should also be noted that salt crystals exhibit a range of roughness values, with a the largest being approximately twice as large as the smallest in our samples, and this range is also seen in the replicas, despite the random choice of particles.

**Table 1.** Roughness parameters of four salt crystals and four salt crystal replica samples.

|  | Salt Crystal | Replica of Salt Crystal |
|---|---|---|
| Area | 0.104 mm$^2$ to 0.255 mm$^2$ | 0.1124 mm to 0.218 mm$^2$ |
| RMS Roughness | 13.27 μm to 26.74μm | 10.63 μm to 22.73μm |
| Mean Roughness | 10.76 μm to 19.16 μm | 8.27 μm to 18.96 μm |
| Skewness | −1.447 to −0.35 | 0.304 to 0.454 |


Also, as a final check to make sure that these values are not significantly affected by the presence of formvar itself, a formvar surface with no crystals present was also scanned and analysed, the RMS roughness in this case being found to be





175 nm. This is orders of magnitude less than the values obtained in Table 1, supporting our assertion that the parameters retrieved from replicas are close to those for the original crystals.

**2.5 Retrieval of Ice Crystal Roughness Parameters**

At this stage, roughness parameters for ice crystals produced in the Manchester Ice Cloud Chamber (MICC) were explored following the procedures described in the methodology section. As a proof of concept, a single temperature (–20°C) was chosen and an ice cloud was produced. In the cloud generation chamber, temperature and humidity are carefully controlled to initiate condensation, allowing water vapor to condense and form ice crystals. These crystals then fall and exit through an

opening at the bottom of the chamber, where they were captured, and replicas were created. The replicas were inspected with and optical profilometer rather than the SEM described for early investigations due to improved speed, access and non-contact aspects of the method.

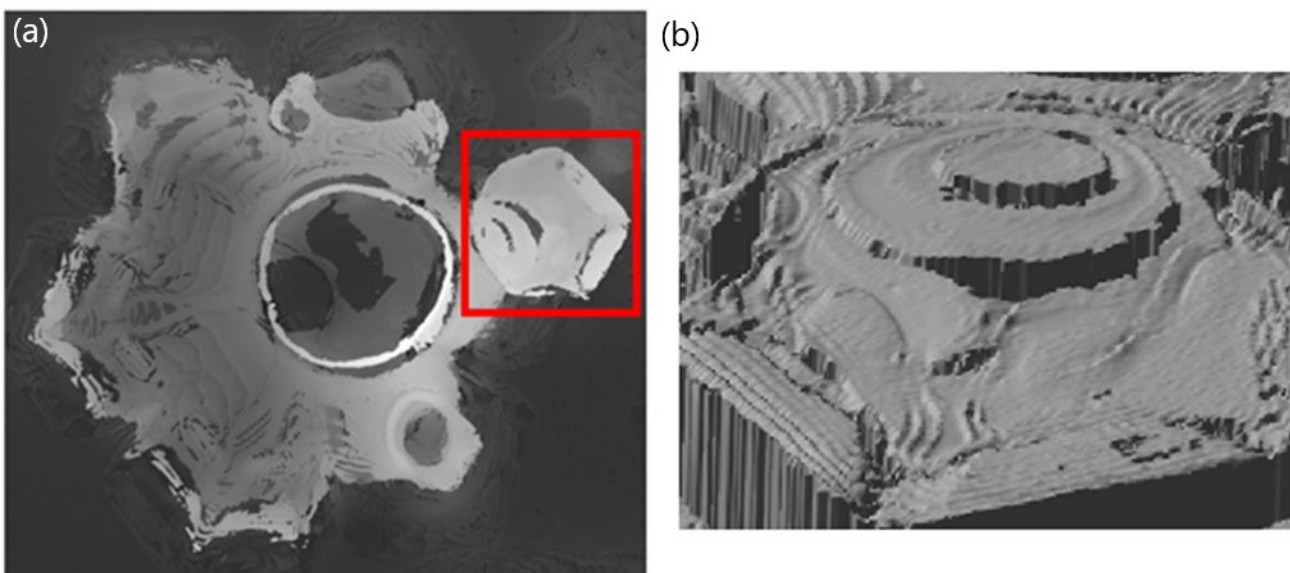

**Figure 8. (a)** One plate sits on another crystal marked with red square. **(b)** Detail of surface scan of crystal highlighted by red square.

An example crystal replica is shown in Figure 8, visually demonstrating that it is possible to capture the detailed surface features and shape of ice crystals with formvar. Multiple ice crystals were analysed using the same software, Gwyddion, to calculate roughness parameters and ascertain a range for the different variables. These results are shown in Table 2.

**Table 2.** Roughness parameters of three ice crystal replicas.

| | Ice Crystal Replica |
| --- | --- |
| Area | $300\ \mu m^2$ to $800\ \mu m^2$ |





| | |
|---|---|
| RMS Roughness | 1.2 μm to 3.6 μm |
| Mean Roughness | 0.8 μm to 2.66 μm |
| Skewness | -2.36 to 0.28 |

Roughness parameters can be compared over 35 areas from four different crystal surface scanning to establish a correlation with the length parameter. By selecting various regions of the ice crystal surface, roughness can be analysed at different scales to provide a more detailed assessment of surface variability. In this study, multiple areas of different length parameter were chosen to capture localized variations, and their roughness values were statistically compared. The results, presented as a box plot in Figure 9, demonstrate the roughness distributions for each selected area, highlighting trends and potential correlations with the length parameter.

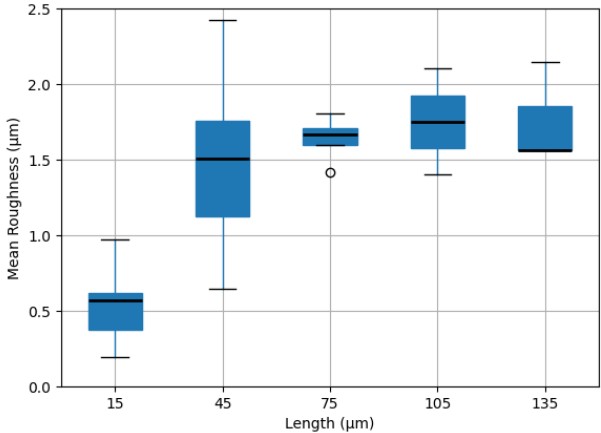

**Figure 9.** Statistical comparison of length (μm) vs. mean-roughness values (μm) across multiple selected areas of multiple ice crystals' surfaces (n=7 for each size bin).

In general, this distribution can be further simplified to show the correlation between length and roughness parameters. As the length increases, larger features on the ice crystal surface correspond to higher roughness values. This relationship reflects the fact that greater lengths encompass larger surface irregularities, leading to higher roughness measurements. The length parameter is derived directly from the size of the selected area, allowing for a clear connection between the dimensions of the area and the resulting roughness values.

As before with the salt crystal analysis a range of roughness values are seen in the samples, though the range is not quite so large for our ice samples, only varying by a factor of 1.5–1.8. However, we acknowledge that the environmental conditions are expected to have a significant effect on the roughness of crystals, particularly the supersaturation of the environment in which they form, and we have presented results from a single set of controlled conditions in developing this technique.





Collier et al. (2016) used the roughness parameters of sand particles as a proxy for model work for asymmetry parameter calculations, yet it is encouraging that findings are relatively close to the parameters in that work. Their two-scale roughness approach was applied to crystals with a size of 32.5 µm, which lies between our lengths 15 and 45 µm. The mean roughness

for the large-scale features was measured at 1.08 µm, while the mean roughness for the small-scale features was measured at 0.13 µm which is approximately the lowest value that we observed with our smallest length scale of 15 µm. In any event roughness will significantly affect the scattering behaviour of the crystals (Collier, et al., 2016), and this is often an underappreciated affect. In simple terms it can explain the relative rare of observance of halos in thin cirrus (Smedley, 2003).

We stress that our study is a proof-of-concept study, and we anticipate that ice crystal roughness will vary with temperature, habit, and supersaturation. This variability is especially significant in high-altitude cirrus clouds and turbulent cumulus clouds, where the irregularities in ice crystals or droplets can lead to increased roughness, significantly impacting light scattering and the overall optical properties of the clouds. The high roughness can lead to higher scattering efficiencies and smaller asymmetry parameters (Ulanowski, et al., 2006), affecting the amount of solar radiation reflected back into space

and complicating satellite retrievals of cloud properties. Capturing these roughness characteristics is essential for improving the reliability of satellite observations and enhancing our understanding of cloud dynamics in climate models.

While this method was developed in a laboratory setting, it could potentially be adapted for airborne campaigns, once certain challenges are overcome. The lower pressure available at altitude will assist rapid enough drying of the formvar, could be

aided by slightly thinner coatings, adapted for the science question at hand. Additionally, this method can be employed at some high-altitude ground-based stations situated at cloud level, allowing for the collection of samples directly from natural clouds. Such efforts are essential, as they can improve our understanding of cloud feedback mechanisms and the intricate details of roughness, ultimately contributing to a more comprehensive understanding of the net energy balance in the atmosphere. Nevertheless, it is important to note that this method depends on the properties of the formvar solution used,

which can introduce variability in the results. Factors such as the concentration of the solution and the conditions under which it is applied can significantly influence the final outcomes, potentially leading to artefacts in the data. However, if appropriate mitigation methods are implemented, the occurrence of these artefacts can be greatly reduced, resulting in the most accurate possible roughness measurements.



## 4 Conclusion

According to the IPCC report (2013), cloud models are inadequate for informing climate change models, which hinders the ability of models to provide accurate climate change predictions. Improvements are necessary in many aspects, especially with regard to their microphysical properties. The microphysical properties directly impact scattering characteristics. By representing the conditions of ice clouds accurately within models, reliability in predictions can be enhanced. While previous studies have extensively examined the shape and habits of ice particles, this study solely concentrates on a method for

roughness parameter retrieval, leaving exploration of the effect of environmental conditions on roughness formation to future investigations.

In calculating roughness parameters, this formvar technique, combined with modern imaging opportunities, offers novel possibilities for direct measurement of the roughness, despite its being based on an older technique. Roughness makes a

critical contribution to scattering properties. Therefore, accurately determining roughness parameters can improve the prediction of scattering outcomes, which in turn can enhance cloud models and make their feedback to climate models more precise. Our method has resulted in the successful creation of ice crystal replicas, and their imaging and analysis at roughness scales by profilometer. The accuracy of the roughness parameters derived was validated through a comparative analysis using salt crystals, where the integrity of the crystals could be maintained independently of the formvar. This

indicated that the roughness parameters found with our methodology have an uncertainty of no more than 5-6%, where some of this uncertainty may be natural variation in the salt crystals.

A critical aspect of this study lies in its innovative integration of established techniques with cutting-edge instruments, which enables the extraction of valuable information on ice crystal roughness parameters. This research provides a detailed method

for capturing and visualizing ice crystals captured in formvar using an optical profilometer, ensuring that key surface characteristics are preserved. Through comprehensive scanning, it allows for the accurate measurement of roughness parameters, which can be incorporated into models to examine their impact on scattering. These findings can be compared with both laboratory data and in-situ measurements, offering new perspectives on cloud microphysics. This work not only enhances our understanding of cloud behaviour but also contributes to refining models that simulate cloud properties, which

are crucial for weather predictions and climate analysis. By bridging the gap between theoretical models and real-world observations, this study plays a key role in advancing our comprehension of cloud dynamics and their influence on the Earth's climate system.




**Code Availability**

The Python code used to in this study is available from the corresponding author upon request.


**Data Availability**

The data used to support the findings of this study are available from the corresponding author upon request.

**Author Contribution**

Omer Celebi conceptualized and designed the study, conducted all data collection, analysis, and interpretation, and drafted the manuscript. Ann R. Webb, Paul Connolly and Andrew R.D. Smedley provided guidance, supervision, and critical revisions of the manuscript.

**Competing interests**

The contact author has declared that none of the authors has any competing interests.

**Acknowledgement**

This research was supported by a scholarship from the Ministry of National Education, Türkiye. I am grateful for their financial support, which made this work possible. The first author wrote the manuscript and improved the language with the
help of generative AI. All other authors, who are native English speakers, reviewed and further improved the language of the manuscript.



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
