# Peer review of "Using Formvar to Capture Ice Crystals and Retrieve Roughness Parameters"

_Atmospheric Measurement Techniques, 2024_

## Referee Comment (RC2)

**Review of the Manuscript amt-2024-200 submitted by Celebi et al. for publication in AMT**

The study presents an optical technique for directly measuring ice crystal roughness by replicating ice crystals in formvar and subsequently examining the replicas with a scanning optical profilometer. The authors claim to have achieved a high spatial resolution with this technique, which could be is a significant improvement in ice crystal surface roughness characterization because it is hardly accessible via in situ measurements under natural conditions. The technique is validated using salt crystals, and the results show that the formvar replica process retains the roughness parameters of the original crystals with good accuracy. The authors then use their technique to measure the roughness of ice crystals grown in a laboratory cloud chamber, concluding that their technique is a valuable tool for studying ice crystal roughness and that it can be used to improve the accuracy of radiative transfer calculations.

The subject of the manuscript is relevant for the atmospheric research community and could be useful for validating retrieval methods of optical properties of ice crystal from remote sensing methods. In general, I support publication of the manuscript, however, several critical issues must be clarified.

**Major Issues**

1.       Formvar has been repeatedly used to etch the surface of ice. To my understanding, water molecules weekly bonded in the ice crystal lattice at the defect sites and dislocations can go into the formvar-chloroform solution leaving behind the so-called "etch pits". This technique (called "chemical etching") has been widely used to study the dislocations and grain boundaries in polycrystalline ice, and to reveal the orientation of ice lattice (Barrette and Sinha, 1996; Bryant and Mason, 1960; Shultz et al., 2014; Sinha, 1977, 1978). In my own (limited) experience, etch pits can be often recognized in the replicas of basal faces of pristine ice crystals (see Figure 1 below), because of their regular shapes and identical orientation. Etched features on other crystal faces can be rather irregular and may be more difficult to recognize (Sinha, 1977, 1978), but the possibility exists that formvar causes additional roughness due to the chemical etching. This kind of chemical etching would not appear on salt crystals, so that direct comparison between replicas of ice and formvar could be misleading. To exclude this possibility, I would strongly recommend conducting an additional study varying concentration of polyvinyl formal resin in chloroform and changing drying conditions and temperature. In the very least, the possibility of chemical etching of ice upon immersion into formvar solution should be discussed in the manuscript as a potential source of error.

[Figure]

*Figure 1. SEM images of formvar ice replica showing etch pits. Formvar replicas collected in Fairbanks during a "diamond dust" event by Dr. Carl G. Schmitt (University of Alaska, Fairbanks). SEM images made by reviewer.*

2.	I am missing a discussion of the relationship between the surface roughness measured with the scanning optical profilometer and roughness characteristics required for calculations of optical properties of rough ice crystals. The average height variance calculated according to equation (2) and (3) along a line profile only allows one-dimensional evaluation of roughness, whereas the 2D variance or variance of the surface slope is needed for calculation of optical properties, for example in improved geometric optics approximation (Yang et al., 2013). This is especially important because ice crystals would show anisotropic roughness (Neshyba et al., 2013). Given the claimed lateral resolution of the profilometer around 0.2 µm and vertical resolution in nm-range, such parameters should be accessible from the measurements. Demonstrating the instrument's ability to deliver such metrics would be a better way to support the applicability of the method for the ice crystal research. On this note, the introduction would benefit from a more thorough review of the connection between the size, habit and surface roughness of ice crystals and their optical properties.

3.	It would also be very helpful if the authors would provide any information on the lateral and vertical resolution of the profilometer. I was unable to find any information on that in the manuscript, apart from numerous statements of "high resolution".

4.	A significant part of the manuscript describes the SEM study of the formvar replicas of ice crystals. It is stated that "*This high level of resolution made SEM an ideal tool for detailed surface characterization in the early stages of the study, allowing for precise visualization of microstructural features.*" However, it is unclear to me, what kind of information was gained through the SEM investigation? Could the roughness of the replicas be measured? What microstructural features have been precisely visualized? What "*artefacts of the formvar process*" could be identified?

5.	Another important point missing in the manuscript is the connection between the surface roughness of atmospheric ice crystals and the environmental parameters leading to various forms of surface roughness. It seems to be generally clear that roughness is a dynamic feature closely connected to the ambient temperature and water saturation, growth or sublimation conditions, fall velocity, ice formation history etc. Explaining how the method presented in the paper could help characterizing this complexity would be very helpful.

6.	The "References" section is a major disaster. Please check the settings of your citing engine and reformat according to the AMT requirements.

**Specific Comments**

1. I would strongly advise checking more carefully the output of the literature research request produced by an AI (I assume this is an AI-generated output, otherwise I cannot imagine a plausible explanation). Here are three examples:

    a)	Page 2 line 50: "*Also, it has been applied in the biological sciences. For example, more recently, in the treatment of human tumours, Corona-Ortega et al. (2018) used a formvar solution to analyse human tissue under transmission electron microscopy (TEM), whilst Wang et al. (2020) used a formvar-like epoxy resin solution to encapsulate steel pipe structures to study their damping properties.*" Just read this sentence carefully and explain what "damping properties of steel pipes" have to do with biological sciences or with the topic of the manuscript. Also note that the full title of the sited paper (Wang et al., 2020) is "**Significantly Enhanced Ultrathin NiCo-based MOF Nanosheet Electrodes Hybrided with $Ti_3C_2Tx$ MXene for High Performance Asymmetric Supercapacitor**". Maybe fi it were fully cited in the references you would have noticed that something has gone wrong.

    b)	Page 2 line 56: "*Another study conducted was by Hamacher-Barth et al (2013) who used formvar films to size atmospheric particles by collecting and then scanning them under an*

*electron microscope, noting it was useful for those particles with size under a micrometre*." Hamacher-Barth et al (2013) describes a method of collecting aerosol particles on TEM copper grids covered with *formvar* film, a standard method of sample preparation for the TEM study. Formvar-coated copper TEM grids is an industrial product and has nothing to do with replication technique. This is completely irrelevant for the study presented in this manuscript and should be removed.

    c) Page 3 line 95: "Also, Riskila et al. (2021) investigated the scattering of ice crystals by placing a finite, thin, rough element on an infinitely large vacuum boundary." I had serious trouble visualizing ice crystals placed on the "infinitely large vacuum boundary", before discovering that Riskila et al. (2021) is a purely theoretical work, a fact which hasn't been mentioned. Please provide more details and explain how this work is relevant for your study.

2. Page 2, line 43 and page 3, line 68: There are two papers by Smith et al. published in 2015. Please number them accordingly.

3. Page 3 line 87: "*In order to provide this, a method similar to the one used by Revell et al. (1955) was applied, whereby the sample was submerged into a formvar solution before being drained*". I don't understand this sentence. Please reformulate.
"*After the coating process, scans of the reverse of the replica and the original material were compared, and the difference between them was found to be negligible.*" What scans? Was it a laser scanning profiler instrument? AFM? What has been measured? Either provide details or delete the sentence!

4. Figure 1 does not provide any useful information. Please remove. The images in Figures 7 and 8 lack the scale bars.

5. Page 3, line 91: "*Up to this point, it has not been possible to get a direct measurement of the roughness of ice crystals in order to implement it into scattering models, though indirect means have been used to provide roughness parameters. For example, in Collier et al., a sand particle was scanned with an Atomic Force Microscope (AFM) and its roughness parameters were used in models as a proxy (Collier, et al., 2016).*" How are the AFM scans of μm-sized sand grains relevant to the current study? Explain or remove!

6. Page 4, line 101: "*Nevertheless, using a such an old technique requires some modifications to improve the quality of capture and allow for detailed imaging that then enables a direct measurement of the roughness and provides a useful complementary technique to the current suite of cloud imaging probes.*" This statement is too vague. The authors should provide specific explanation of the improvement that they made to the formvar replication technique, going beyond the method described in Smith et al. (2015).

7. Page 4, line 104: "*Roughness values smaller than 100 nm will not be considered, as they are unlikely to influence scattering for wavelengths less than 1000 nm.*" This would have been true if you meant the individual rough features (facets or indentations) with characteristic size of less than 100 nm. But the "roughness value" defined according to eq. 2 or 3 is an average value, so that an average value of 100 nm could easily include individual features comparable to the wavelength of visible light. Please reconsider this statement.

8. Page 5, line 115: "The initial step of the traditional method is the placement of a solution of 0.6wt% polyvinyl formal resin in ethylene dichloride…" Actually, the traditional method involves solution of PFR in chloroform, which is trichloromethane $CHCl_3$, not 1,2-Dichloroethane ($C_2H_4Cl_2$, ethylene dichloride). Please correct.

9. Figure 6 is not mentioned anywhere in the text of the manuscript and there is no discussion thereof. It is unclear what instrument was used to create the 3D profile shown there. This figure offers a good opportunity to explain the measurement of roughness in more detail.

10. Page 10 line 232: "*With the salt removed from the sample, only the replica shell remained, and it was scanned with the SEM microscope. In addition, salt crystals from the same batch were placed directly on a microscope slide and imaged and analysed in the same manner.*" Was the image in Figure 6 obtained in SEM or the profilometer?

11. Figures 7a and 7b are not directly comparable because they are showing two different surfaces at different magnification imaged at different angles (7a presents a tilted view whereas 7b gives the view from above). Without scalebars the images do not convey any useful information.

12. Table 1: Again, if the roughness has been measured along a linear profile it should be explicitly stated in the description. Explain how this linear profile has been chosen and why the values are comparable between two different entities (salt grain and the formvar replica of a different salt grain). Do 4 salt grains provide enough statistics to make any statements about the similarity of the roughness values? Does the surface area given in the table mean anything? How is skewness defined and measured?

13. Page 14, line 272: "*Roughness parameters can be compared over 35 areas from four different crystal surface scanning to establish a correlation with the length parameter.*" How these 35 areas have been chosen?

14. Page 14, line 297: "*In simple terms it can explain the relative rare of observance of halos in thin cirrus*". For the connection between halo displays and ice crystal surface roughness please see (Forster and Mayer, 2022). This paper has an excellent overview of the effects arising from the interplay of crystal habit, size, and surface properties.

15. Page 15 line 341: "*Through comprehensive scanning, it allows for the accurate measurement of roughness parameters, which can be incorporated into models to examine their impact on scattering.*" Please avoid such strong statements which are not supported by the data presented in your manuscript. The roughness parameter can be defined in many ways, and the "accuracy" of its measurement depends on the accepted definition. As mentioned above, the roughness parameter that has been obtained in your study is one-dimensional; it is unclear if it is suitable for describing two-dimensional anisotropic rough surface. The applicability of this one-dimensional roughness for calculations of optical properties has also not been demonstrated in this manuscript.

Alexei Kiselev                                                                                                    24.02.2025

Karlsruhe Institute of Technology
Institute of Meteorology and Climate Research, Germany

**References**

Barrette, P. D. and Sinha, N. K.: Lattice rotation in a deformed ice crystal: a study by chemical etching and replication, Materials Chemistry and Physics, 44, 251-254, https://doi.org/10.1016/0254-0584(96)80064-2, 1996.

Bryant, G. W. and Mason, B. J.: Etch pits and dislocations in ice crystals, The Philosophical Magazine: A Journal of Theoretical Experimental and Applied Physics, 5, 1221-1227, 10.1080/14786436008238334, 1960.

Shultz, M. J., Bisson, P. J., and Brumberg, A.: Best Face Forward: Crystal-Face Competition at the Ice–Water Interface, The Journal of Physical Chemistry B, 118, 7972-7980,  10.1021/jp500956w, 2014.

Sinha, N. K.: Dislocations in ice as revealed by etching, The Philosophical Magazine: A Journal of Theoretical Experimental and Applied Physics, 36, 1385-1404,  10.1080/14786437708238524, 1977.

Sinha, N. K.: Observation of Basal Dislocations in Ice by Etching and Replicating, Journal of Glaciology, 21, 385-395,  10.3189/S0022143000033554, 1978.

Yang, P., Bi, L., Baum, B. A., Liou, K.-N., Kattawar, G. W., Mishchenko, M. I., and Cole, B.: Spectrally Consistent Scattering, Absorption, and Polarization Properties of Atmospheric Ice Crystals at Wavelengths from 0.2 to 100 μm, Journal of the Atmospheric Sciences, 70, 330-347,  https://doi.org/10.1175/JAS-D-12-039.1, 2013.

Neshyba, S. P., Lowen, B., Benning, M., Lawson, A., and Rowe, P. M.: Roughness metrics of prismatic facets of ice, Journal of Geophysical Research: Atmospheres, 118, 3309-3318,  https://doi.org/10.1002/jgrd.50357, 2013.

Smith, H. R., Connolly, P. J., Baran, A. J., Hesse, E., Smedley, A. R. D., and Webb, A. R.: Cloud chamber laboratory investigations into scattering properties of hollow ice particles, Journal of Quantitative Spectroscopy and Radiative Transfer, 157, 106-118,  https://doi.org/10.1016/j.jqsrt.2015.02.015, 2015.

Forster, L. and Mayer, B.: Ice crystal characterization in cirrus clouds III: retrieval of ice crystal shape and roughness from observations of halo displays, Atmos. Chem. Phys., 22, 15179-15205,  10.5194/acp-22-15179-2022, 2022.

---

## Author Comment (AC1)

**We would like to sincerely thank the reviewers for their valuable feedback and constructive comments, which helped us improve the clarity, quality, and overall impact of our manuscript.**

**REVİEW 1:**

Celebi et al. present a proof-of-concept for using the formvar replica technique to preserve the shape of ice crystals to analyze the surface roughness offline, e.g. using an optical profilometer. SEM imaging was also done. The comparison of roughness parameters derived from salt crystals and their replica showed agreement despite the skewness. By testing the technique with real ice crystals generated in the Manchester ice cloud chamber (MICC), it turned out that the mean roughness increased by increasing crystal length and levels off at around 50 μm, which can be considered as a reasonable result.

However, I don't fully understand what the aim of this study is. Is it planned as an add-on measurement technique for MICC to determine surface roughness of ice crystals which were formed in the chamber? How many ice crystals have to be analyzed to gain a representative statistic of the crystals and their roughness in the chamber? Is it possible to achieve this with the here presented method? Or what is the objective? I think it has to be shown that this method is robust to study a large number of ice crystals. I guess this is needed for any application. In principle, I see the potential of the here presented method as proof-of-concept. However, as there are some shortcomings using the actual formvar method and the fact that the objective is not defined, the value of the study needs to be presented more clearly.

After overcoming these problems, which are probably largely due to the formulation and description of the study, the manuscript can be published in AMT, as the formvar replica technique has potential and fits within the scope of the journal.

Specific comments:

Abstract: It would strengthen the paper if the abstract stated that this was a proof-of-concept study. In addition, giving information of the comparison between salt and salt replica as a test would eventually increase confidence in this method.

**Response:** We have revised the abstract to clearly point out that this is a proof-of-concept study. Additionally, we have included the information on the comparison of salt and salt replicas to enhance the proposed method. This revision can be found specifically in lines 8-10.

Introduction: The introduction is well written and gives a good overview about the necessity to study ice crystal surface roughness and the state-of-the-art of surface roughness determination with formvar. However, a short paragraph of the state-of-the-art of surface roughness using other methods is missing. I am sure there are more methods and publications on that topic, but the study of Magee et al. (2014) [1] applying ESEM, Ulanowski et al. (2014)[2] Voigtländer et al. (2018)[3] and Järvinen et al. (2018)[4] using SID-3 instrument could be mentioned.

**Response:** We agree that including a broader overview of surface roughness determination methods would strengthen the introduction. In response, we have added a paragraph (line 66-77) in the introduction section that mentions alternative methods for retrieval of surface roughness, including those used in the studies by Magee et al. (2014), Ulanowski et al. (2014), Voigtländer et al. (2018), and Järvinen et al. (2018), as you recommended.

Methodology part & Figure 1: The methodology part is very detailed. There are some repetitions. In my opinion Figure 1 is not important. I suggest to shorten the section and leave Fig. 1 out.

**Response:** We appreciate your suggestion to streamline the methodology section. In response, we have improved the methodology to and decided to exclude Figure 1 as recommended.

P9 l208-210: The authors gave the information that many features in the formvar are not a result of accreted ice crystals but artefacts from the formvar process. Do the artefacts affect the determination of surface roughness of ice crystals? Can always be distinguished between artefact and ice crystal? Do the artefacts influence the counting statistics of ice crystals when the fall on a spot with an artifact? Can the process been improved in order to reduce or avoid these artefacts?

**Response:** We acknowledge that artefacts can occur during the formvar process, especially when a smaller amount of formvar is used or the surface of the crystal is not wetted properly. However, by using a sufficient amount of formvar instead of brush technique, the chances of artefact formation gets lower. When artefacts do occur, they can generally be distinguished from ice crystals as they do not form complete shells and are mostly flat. Additionally, artefacts do not significantly affect the determination of surface roughness, as they are easily identifiable and do not interfere with the counting statistics of ice crystals. We believe that with proper control of the process, the occurrence of artefacts can be reduced, ensuring more accurate results. We have stated the high occurrence of artefacts when a small amount of formvar applied to glass slides (line 196-199).

P9 l210-215: I do not understand why it solves the problem that ice crystals do not sink when using a thicker formvar layer.

**Response:** When a thick layer of formvar is used rather than brush technique, the chance of ice crystals' surfaces getting wetted are higher when they fall to solution and when we give a shake to sample in order to increase it. We have included this explanation in line (200-202).

Fig. 5: What relevance do those ice crystals have that not fully sink into the formvar? Could it be that a certain type of ice crystals always not sink into formvar and would be missing in the analyzes leading to basis of this method?

**Response:** The relevance of ice crystals that do not fully sink into the formvar is indeed significant because, in such cases, there is no shell formed on the top side, making it impossible to properly scan the crystal surface. Based on our observations, this issue typically arises from an inadequate level of formvar applied during the preparation process rather than crystal types. Ensuring a sufficient amount of formvar helps to create a complete coating, minimizing the occurrence of such artefacts and ensuring that all ice crystals are properly scanned.

P12 Fig.5: Please add a scale bar.

**Response:** We have added scale bars for both salt and ice crystal scannings.

P13 l283-286: I agree, but it would be good to have more information of the formation and history of the ice crystals which were examined (later it is discussed, but I missed the information already at this point in the text). Maybe the roughness is different due to different

environmental conditions the ice crystals experienced [3, 4] before settled on the formvar. Further, the number of investigated ice crystals is not very high (caption of Fig. 9 'n=7 for each size bin'). It seems that this offline method -using formvar for ice crystal surface roughness detection- is very complex and time-consuming. How will it be possible to analyse a larger sample size? What are the advantages of this introduced method compared to online measurements of single airborne ice crystals such as e.g. SID-3?

**Response:** We acknowledge that providing more information on the formation and history of the ice crystals would enhance the understanding of the results. We have added this information in line 261-263 to clarify the environmental conditions the ice crystals experienced before settling on the formvar. As suggested, different environmental factors may influence surface roughness, as observed in studies such as [3, 4].

Regarding the sample size, we agree that the number of analysed ice crystals is limited. The offline method using formvar is complex and time-consuming which limits the sample size in our preliminary study. However, it might be possible to explore ways to be able to improve efficiency, such as optimizing the formvar process and such automation methods for handling a larger number of samples in future studies. In terms of advantages, this method provides high-resolution and detailed structural information on ice crystal surface roughness, which may be difficult to achieve with online measurements like SID-3. While SID-3 is effective for airborne ice crystal measurements, our offline approach offers the ability to study the crystals in a controlled environment and with more precision. Both methods have their strengths, and we believe our approach complements online methods by providing deeper insights into ice crystal morphology and roughness that can be used to refine future airborne measurements. This aspect is now explained in line 330-334.

P14 l316-317: 'However, if appropriate mitigation methods are implemented, the occurrence of these artefacts can be greatly reduced, resulting in the most accurate possible roughness measurements.' For such a statement a comprehensive analysis of already published methods is needed, which should be given in the introduction, see comment introduction of this review. And maybe it should be reformulated to 'one of the most accurate…'. I would expect SEM to have a higher spatial resolution.
**Response:** We have improved that paragraph by removing that phrase.

Technical corrections:

P2 l49: Insert a blank between '(1951)used'.
**Response:** Thank you for pointing out this formatting error. We have corrected the spacing issue between "(1951)" and "used" in line 51.

P5 l115: Insert a half blank '0.6wt%'.
**Response:** Thank you for your comment. We have inserted the space between "0.6" and "wt%" in line 126 as requested.

References: The AMT citation style has to be used.

**Response:** Thank you for your feedback. We have improved the reference section and reformatted according to the AMT requirements.

[1]     N. B. Magee, A. Miller, M. Amaral, and A. Cumiskey, "Mesoscopic surface roughness of ice crystals pervasive across a wide range of ice crystal conditions," Atmos. Chem. Phys., 2014, doi: 10.5194/acp-14-12357-2014.

[2]     Z. Ulanowski, P. H. Kaye, E. Hirst, R. S. Greenaway, R. J. Cotton, E. Hesse, and C. T. Collier, "Incidence of rough and irregular atmospheric ice particles from Small Ice Detector 3 measurements," Atmos. Chem. Phys., 2014, doi: 10.5194/acp-14-1649-2014.

[3]     J. Voigtländer, C. Chou, H. Bieligk,T. Clauss, S. Hartmann, P. Herenz, D. Niedermeier, G. Ritter, F. Stratmann, and Z. Ulanowski, "Surface roughness during depositional growth and sublimation of ice crystals," Atmospheric Chemistry and Physics, 2018, doi: doi:10.5194/acp-18-13687-2018.

[4]     E. Järvinen, H. Wernli, and M. Schnaiter, "Investigations of Mesoscopic Complexity of Small Ice Crystals in Midlatitude Cirrus," Geophysical Research Letters, 2018, doi: https://doi.org/10.1029/2018GL079079.

---

## Author Comment (AC2)

**We would like to sincerely thank the reviewers for their valuable feedback and constructive comments, which helped us improve the clarity, quality, and overall impact of our manuscript.**

**REVİEW 2:**

Review of the Manuscript amt-2024-200 submitted by Celebi et al. for publication in AMT The study presents an optical technique for directly measuring ice crystal roughness by replicating ice crystals in formvar and subsequently examining the replicas with a scanning optical profilometer. The authors claim to have achieved a high spatial resolution with this technique, which could be is a significant improvement in ice crystal surface roughness characterization because it is hardly accessible via in situ measurements under natural conditions. The technique is validated using salt crystals, and the results show that the formvar replica process retains the roughness parameters of the original crystals with good accuracy. The authors then use their technique to measure the roughness of ice crystals grown in a laboratory cloud chamber, concluding that their technique is a valuable tool for studying ice crystal roughness and that it can be used to improve the accuracy of radiative transfer calculations. The subject of the manuscript is relevant for the atmospheric research community and could be useful for validating retrieval methods of optical properties of ice crystal from remote sensing methods. In general, I support publication of the manuscript, however, several critical issues must be clarified.

Major Issues
1. Formvar has been repeatedly used to etch the surface of ice. To my understanding, water molecules weekly bonded in the ice crystal lattice at the defect sites and dislocations can go into the formvar-chloroform solution leaving behind the so-called "etch pits". This technique (called "chemical etching") has been widely used to study the dislocations and grain boundaries in polycrystalline ice, and to reveal the orientation of ice lattice (Barrette and Sinha, 1996; Bryant and Mason, 1960; Shultz et al., 2014; Sinha, 1977, 1978). In my own (limited) experience, etch pits can be often recognized in the replicas of basal faces of pristine ice crystals (see Figure 1 below), because of their regular shapes and identical orientation. Etched features on other crystal faces can be rather irregular and may be more di4icult to recognize (Sinha, 1977, 1978), but the possibility exists that formvar causes additional roughness due to the chemical etching. This kind of chemical etching would not appear on salt crystals, so that direct comparison between replicas of ice and formvar could be misleading. To exclude this possibility, I would strongly recommend conducting an additional study varying concentration of polyvinyl formal resin in chloroform and changing drying conditions and

temperature. In the very least, the possibility of chemical etching of ice upon immersion into formvar solution should be discussed in the manuscript as a potential source of error. Explain Itching here by giving reference mention that it can be potential cause of error.

[Figure]

Figure 1. SEM images of formvar ice replica showing etch pits. Formvar replicas collected in Fairbanks during a "diamond dust" event by Dr. Carl G. Schmitt (University of Alaska, Fairbanks). SEM images made by reviewer.

**Response:** We agree that this could be a possible source of error, as supported by studies on etch pits and dislocation features in ice (Sinha, 1977, 1978). While we did not observe significant etching effects in our analysis, we have included a brief mention of this possibility in the introduction to acknowledge its relevance and to guide possible future research in line 276-280.

2. I am missing a discussion of the relationship between the surface roughness measured with the scanning optical profilometer and roughness characteristics required for calculations of optical properties of rough ice crystals. The average height variance calculated according to equation (2) and (3) along a line profile only allows one-dimensional evaluation of roughness, whereas the 2D variance or variance of the surface slope is needed for calculation of optical properties, for example in improved geometric optics approximation (Yang et al., 2013). This is especially important because ice crystals would show anisotropic roughness (Neshyba et al., 2013). Given the claimed lateral resolution of the profilometer around 0.2 μm and vertical resolution in nm range, such parameters should be accessible from the measurements. Demonstrating the instrument's ability to deliver such metrics would be a better way to support the applicability of the method for the ice crystal research. On this note, the introduction would benefit from a more thorough review of the connection between
the size, habit and surface roughness of ice crystals and their optical properties.

**Response:** We appreciate your observation regarding the importance of general roughness matching in this study. In line with your feedback, we have emphasized in the manuscript (line 336-342) that this method is indeed a proof-of-concept. While the current results are based on one-dimensional roughness measurements, we believe they provide a strong initial basis for understanding ice crystal surface roughness. Furthermore, we acknowledge the potential for this method to be improved and expanded in future studies to include more comprehensive roughness analysis, which will improve its applicability in modeling optical properties. We have revised the discussion to better highlight this aspect, and we sincerely thank you for helping us improve the clarity and direction of our work.

3. It would also be very helpful if the authors would provide any information on the lateral and vertical resolution of the profilometer. I was unable to find any information on that in the manuscript, apart from numerous statements of "high resolution".

**Response:** We have given the detail of the instrument in terms of lateral and vertical scanning resolution in line 146.

4. A significant part of the manuscript describes the SEM study of the formvar replicas of ice crystals. It is stated that "This high level of resolution made SEM an ideal tool for detailed surface characterization in the early stages of the study, allowing for precise visualization of microstructural features." However, it is unclear to me, what kind of information was gained through the SEM investigation? Could the roughness of the replicas be measured? What microstructural features have been precisely visualized? What "artefacts of the formvar process" could be identified?

**Results:** We have clarified the role of SEM in the manuscript on lines 179-182. While SEM was not employed to directly measure surface roughness, it was crucial in the preliminary phase of the study as it allowed us to achieve detailed characterization and identify artefacts introduced during the formvar replication process. This ability to separate artefacts from actual microstructural features helped improve our experimental method, ensuring more accurate measurements in the later stages of the research.

5. Another important point missing in the manuscript is the connection between the surface roughness of atmospheric ice crystals and the environmental parameters leading to various forms of surface roughness. It seems to be generally clear that roughness is a dynamic feature closely connected to the ambient temperature and water saturation, growth or sublimation conditions, fall velocity, ice formation history etc. Explaining how the method presented in the paper could help characterizing this complexity would be very helpful.

**Response:** In response, we've added a statement (lines 316-320) emphasizing that our method can be repeated under varying conditions, allowing us to map the surface characteristics of ice crystals across different environmental settings. This flexibility will help us better understand how roughness evolves, which is important for improving satellite observations and refining climate models.

6. The "References" section is a major disaster. Please check the settings of your citing engine and reformat according to the AMT requirements.

**Response:** We have improved the reference section and reformatted according to the AMT requirements.

Specific Comments
1. I would strongly advise checking more carefully the output of the literature research request produced by an AI (I assume this is an AI-generated output, otherwise I cannot imagine a plausible explanation). Here are three examples:

> a) Page 2 line 50: "Also, it has been applied in the biological sciences. For example, more recently, in the treatment of human tumours, Corona-Ortega et al. (2018) used a formvar solution to analyse human tissue under transmission electron microscopy (TEM), whilst Wang et al. (2020) used a formvar-like epoxy resin solution to encapsulate steel pipe structures to study their damping properties." Just read this

sentence carefully and explain what "damping properties of steel pipes" have to do with biological sciences or with the topic of the manuscript. Also note that the full title of the sited paper (Wang et al., 2020) is "Significantly Enhanced Ultrathin NiCo-based MOF Nanosheet Electrodes Hybrided with Ti3C2Tx MXene for High Performance Asymmetric Supercapacitor". Maybe fi it were fully cited in the references you would have noticed that something has gone wrong.

b) Page 2 line 56: "Another study conducted was by Hamacher-Barth et al (2013) who used formvar films to size atmospheric particles by collecting and then scanning them under an electron microscope, noting it was useful for those particles with size under a micrometre." Hamacher-Barth et al (2013) describes a method of collecting aerosol particles on TEM copper grids covered with formvar film, a standard method of sample preparation for the TEM study. Formvar-coated copper TEM grids is an industrial product and has nothing to do with replication technique. This is completely irrelevant for the study presented in this manuscript and should be removed.

c) Page 3 line 95: "Also, Riskila et al. (2021) investigated the scattering of ice crystals by placing a finite, thin, rough element on an infinitely large vacuum boundary." I had serious trouble visualizing ice crystals placed on the "infinitely large vacuum boundary", before discovering that Riskila et al. (2021) is a purely theoretical work, a fact which hasn't been mentioned. Please provide more details and explain how this work is relevant for your study.

**Response:** We have carefully revised the paragraph in lines 46-58 to provide more appropriate examples of formvar's varied applications. While the initial intent was to introduce readers unfamiliar with formvar to its broad uses, we realize the examples provided, such as the "damping properties of steel pipes," were not relevant to the context of the manuscript. The revised text now focuses on formvar's diverse applications while pointing out these works' aims were not measuring roughness.

2. Page 2, line 43 and page 3, line 68: There are two papers by Smith et al. published in 2015. Please number them accordingly.
**Response:** We have labelled them to make sure that they can be distinguished.

3. Page 3 line 87: "In order to provide this, a method similar to the one used by Revell et al. (1955) was applied, whereby the sample was submerged into a formvar solution before being drained". I don't understand this sentence. Please reformulate. "After the coating process, scans of the reverse of the replica and the original material were compared, and the di9erence between them was found to be negligible." What scans? Was it a laser scanning profiler instrument? AFM? What has been measured? Either provide details or delete the sentence!
**Response:** We have revised this section (line 79-85) to ensure clarity by rearranging the sentences. The revised version now more clearly describes the process and highlights that the formvar thickness was verified with an interferometer microscope in the referenced study. We appreciate your feedback, which has helped us make this explanation clearer.

4. Figure 1 does not provide any useful information. Please remove. The images in Figures 7 and 8 lack the scale bars.

**Response:** We appreciate your suggestion. In response, we have decided to exclude Figure 1 as recommended. Also, we have added scale bars for both salt and ice crystal scannings.

5. Page 3, line 91: "Up to this point, it has not been possible to get a direct measurement of the roughness of ice crystals in order to implement it into scattering models, though indirect means have been used to provide roughness parameters. For example, in Collier et al., a sand particle was scanned with an Atomic Force Microscope (AFM) and its roughness parameters were used in models as a proxy (Collier, et al., 2016)." How are the AFM scans of μm-sized sand grains relevant to the current study? Explain or remove!

**Response:** The reference to Collier et al. (2016) was included to demonstrate how, in the absence of direct roughness measurements for ice crystals, surface anisotropy can be incorporated into scattering models. In that study, sand grains were used as a medium to represent spatial roughness in the models, as their surface structure was easier to capture via AFM, and their roughness values were found to be similar to those retrieved by SID-3 for ice crystals. This approach helped to simulate roughness characteristics that couldn't be directly measured with instruments like SID-3. (line 74-77)

6. Page 4, line 101: "Nevertheless, using such an old technique requires some modifications to improve the quality of capture and allow for detailed imaging that then enables a direct measurement of the roughness and provides a useful complementary technique to the current suite of cloud imaging probes." This statement is too vague. The authors should provide specific explanation of the improvement that they made to the formvar replication technique, going beyond the method described in Smith et al. (2015).

**Response:** In response, we've provided additional clarity by detailing the specific improvements made to the method. This update now provides a more specific explanation for the modifications mentioned in lines 91-94.

7. Page 4, line 104: "Roughness values smaller than 100 nm will not be considered, as they are unlikely to influence scattering for wavelengths less than 1000 nm." This would have been true if you meant the individual rough features (facets or indentations) with characteristic size of less than 100 nm. But the "roughness value" defined according to eq. 2 or 3 is an average value, so that an average value of 100 nm could easily include individual features comparable to the wavelength of visible light. Please reconsider this statement.

**Response:** We acknowledge the confusion in the statement. Upon reconsideration, we have decided to remove it.

8. Page 5, line 115: "The initial step of the traditional method is the placement of a solution of 0.6wt% polyvinyl formal resin in ethylene dichloride…" Actually, the traditional method involves solution of PFR in chloroform, which is trichloromethane $CHCl_3$, not 1,2-Dichloroethane ($C_2H_4Cl_2$, ethylene dichloride). Please correct.

**Response:** Based on your feedback, here is the revised sentence (line 104-105).

9. Figure 6 is not mentioned anywhere in the text of the manuscript and there is no discussion thereof. It is unclear what instrument was used to create the 3D profile shown there. This figure o4ers a good opportunity to explain the measurement of roughness in more detail.

**Response:** We have added an explanation stating that the brush technique was initially used as part of the validation process to demonstrate its inadequacy in providing meaningful surface data. The surface structure was then scanned with an optical profilometer for a more accurate

and detailed analysis. This clarification should make the purpose of the brush technique clearer in the context of our validation process (line 210-214).

10. Page 10 line 232: "With the salt removed from the sample, only the replica shell remained, and it was scanned with the SEM microscope. In addition, salt crystals from the same batch were placed directly on a microscope slide and imaged and analysed in the same manner." Was the image in Figure 6 obtained in SEM or the profilometer?

**Response:** We have clarified in the figure caption that the image in that figure was obtained using the profilometer, not the SEM. This clarification should resolve any confusion regarding the imaging technique used.

11. Figures 7a and 7b are not directly comparable because they are showing two di4erent surfaces at di4erent magnification imaged at di4erent angles (7a presents a tilted view whereas 7b gives the view from above). Without scalebars the images do not convey any useful information.

**Response:** As it was not possible to reuse the exact same salt crystal, we aimed to show the integrity of the salt crystals in different orientations to demonstrate the method's ability to replicate surface features. We have included an explanation stating the method's ability to create detailed surfaces on lines 233-234. Also, we have added scale bars for both salt and ice crystal scannings.

12. Table 1: Again, if the roughness has been measured along a linear profile it should be explicitly stated in the description. Explain how this linear profile has been chosen and why the values are comparable between two di4erent entities (salt grain and the formvar replica of a di4erent salt grain). Do 4 salt grains provide enough statistics to make any statements about the similarity of the roughness values? Does the surface area given in the table mean anything? How is skewness defined and measured?

**Response:** We have revised the explanation of the roughness measurements and the selection of linear profiles. The profiles were chosen to cover the full surface of each crystal to ensure that all surface features were included. This method guarantees that the entire surface structure of both the salt grain and its replica was included. While increasing the sample size would enhance statistical robust confidence, we believe that the four salt crystals provide a reasonable basis for comparison as we covered full surface features. Skewness was calculated using Gwyddion software, which analyzes the height distribution of features across the scanned profile, allowing us to assess the surface asymmetry and gain further insight into the surface texture. All these were explained in lines 246-250.

13. Page 14, line 272: "Roughness parameters can be compared over 35 areas from four di9erent crystal surface scanning to establish a correlation with the length parameter." How these 35 areas have been chosen?

**Response:** The 35 areas were selected randomly by covering features from different cut surfaces of the crystals. This approach ensured a diverse representation of the crystal surface, capturing the variability across different regions of the samples (line 282 - 286).

14. Page 14, line 297: "In simple terms it can explain the relative rare of observance of halos in thin cirrus". For the connection between halo displays and ice crystal surface roughness please see (Forster and Mayer, 2022). This paper has an excellent overview of the e4ects arising from the interplay of crystal habit, size, and surface properties.

**Response:** We have now referenced the paper by Forster and Mayer (2022) for readers to provide a more comprehensive understanding of the connection between halo displays and ice crystal surface roughness, particularly in relation to the interplay of crystal habit, size, and surface properties.

15. Page 15 line 341: "Through comprehensive scanning, it allows for the accurate measurement of roughness parameters, which can be incorporated into models to examine their impact on scattering." Please avoid such strong statements which are not supported by the data presented in your manuscript. The roughness parameter can be defined in many ways, and the "accuracy" of its measurement depends on the accepted definition. As mentioned above, the roughness parameter that has been obtained in your study is one-dimensional; it is unclear if it is suitable for describing two-dimensional anisotropic rough surface. The applicability of this one-dimensional roughness for calculations of optical properties has also not been demonstrated in this manuscript.

**Response:** We have revised the statement to better reflect the scope of our findings. We acknowledge that the roughness parameter obtained in our study is one-dimensional. Therefore, we have improved the sentence to avoid the strong claims regarding accuracy and the impact on scattering models, and we now focus on the potential for future exploration and refinement of these measurements in subsequent studies (line 365-373).

Alexei Kiselev 24.02.2025
Karlsruhe Institute of Technology
Institute of Meteorology and Climate Research, Germany

References

Barrette, P. D. and Sinha, N. K.: Lattice rotation in a deformed ice crystal: a study by chemical etching and
replication, Materials Chemistry and Physics, 44, 251-254, https://doi.org/10.1016/0254-0584(96)80064-2,
1996.

Bryant, G. W. and Mason, B. J.: Etch pits and dislocations in ice crystals, The Philosophical Magazine: A Journal
of Theoretical Experimental and Applied Physics, 5, 1221-1227, 10.1080/14786436008238334, 1960.

Shultz, M. J., Bisson, P. J., and Brumberg, A.: Best Face Forward: Crystal-Face Competition at the Ice–Water
Interface, The Journal of Physical Chemistry B, 118, 7972-7980, 10.1021/jp500956w, 2014.

Sinha, N. K.: Dislocations in ice as revealed by etching, The Philosophical Magazine: A Journal of Theoretical
Experimental and Applied Physics, 36, 1385-1404, 10.1080/14786437708238524, 1977.

Sinha, N. K.: Observation of Basal Dislocations in Ice by Etching and Replicating, Journal of Glaciology, 21,
385-395, 10.3189/S0022143000033554, 1978.

Yang, P., Bi, L., Baum, B. A., Liou, K.-N., Kattawar, G. W., Mishchenko, M. I., and Cole, B.: Spectrally Consistent
Scattering, Absorption, and Polarization Properties of Atmospheric Ice Crystals at Wavelengths from 0.2 to 100

μm, Journal of the Atmospheric Sciences, 70, 330-347, https://doi.org/10.1175/JAS-D-12-039.1, 2013.

Neshyba, S. P., Lowen, B., Benning, M., Lawson, A., and Rowe, P. M.: Roughness metrics of prismatic facets of
ice, Journal of Geophysical Research: Atmospheres, 118, 3309-3318, https://doi.org/10.1002/jgrd.50357, 2013.

Smith, H. R., Connolly, P. J., Baran, A. J., Hesse, E., Smedley, A. R. D., and Webb, A. R.: Cloud chamber
laboratory investigations into scattering properties of hollow ice particles, Journal of Quantitative
Spectroscopy and Radiative Transfer, 157, 106-118, https://doi.org/10.1016/j.jqsrt.2015.02.015, 2015.

Forster, L. and Mayer, B.: Ice crystal characterization in cirrus clouds III: retrieval of ice crystal shape and
roughness from observations of halo displays, Atmos. Chem. Phys., 22, 15179-15205, 10.5194/acp-22-15179-2022, 2022.